# communications
# earth & environment

# Satellite forecasting of crop harvest can trigger a cross-hemispheric production response and improve global food security

Tetsuji Tanaka[1,2], Laixiang Sun [2,3✉], Inbal Becker-Reshef[2], Xiao-Peng Song [2] & Estefania Puricelli[2]

Global food security is increasingly threatened by climate change and regional human conflicts. Abnormal fluctuations in crop production in major exporting countries can cause volatility in food prices and household consumption in importing countries. Here we show that timely forecasting of crop harvest from satellite data over major exporting regions can trigger production response in the opposite hemisphere to offset the short-term fluctuations and stabilize global food supply. Satellite forecasting can reduce the fluctuation extents of country-level prices by 1.1 to 12.5 percentage points for anticipated wheat shortage or surplus in Russia and Ukraine, and even reverse the price shock in importing countries for anticipated soybean shortage in Brazil. Our research demonstrates that by leveraging the seasonal lags in crop calendars between the Northern and Southern Hemispheres, operational crop monitoring from satellite data can provide a mechanism to improve global food security.

[1] Department of Economics, Meiji Gakuin University, Tokyo, Japan. [2] Department of Geographical Sciences, University of Maryland, College Park, MD, USA. [3] School of Finance & Management, SOAS University of London, Russell Square, London, UK. ✉email: LSun123@umd.edu

The global food crisis resulted from the COVID-19 pandemic and the Russian war on Ukraine illustrates the fragility of our current food system[1,2]. These unpredicted events exaggerate the well-anticipated shocks to food production from climate change[3]. Many cases of shortages in food production caused by extreme weather events or human conflicts are over the regional scale but the impacts could propagate across the globe to trigger widespread crises. In particular, global trade could bring high food prices to the most vulnerable locations, which may be different from the locations directly affected by severe weather e.g., drought[4–7]. Thus, enhancing the resilience of the global food system is a paramount task facing the world today.

Earth observations from satellites are transforming global agriculture monitoring. One of the unique advantages of Earth observations is their synoptic view of environmental change provided increasingly in a timely manner. Recent progress in the assessment of crop growing conditions using satellite remote-sensing data and techniques provides a promising way to alleviate some of the adverse impacts of severe drought on the world's food market. Advances in this field have allowed us to make significant strides in estimating cereal yields at a national scale, with a level of accuracy surpassing alternative forecasting methods, and providing this information one to two months prior to harvest[8–11]. This improved capability of satellite-based crop forecasting at broad scales opens opportunities to enhance the stability of food production, supply and price at the global scale. Intuitively, crop calendars are complementary with each other between the Northern and Southern Hemispheres (Table 1). Therefore, reliable and timely information on crop harvest in one hemisphere could potentially stimulate or suppress production in the opposite hemisphere by enabling farmers to leverage the seasonal lags in crop calendars. In the business world, it is reported that soybean and corn producers in Brazil frequently monitor developments in the US Corn Belt and the US farmers do the same to the Southern Hemisphere[12]. Such a cross-hemispheric response could mitigate price volatilities in world agricultural markets. The recent literature reviews of Pearlman et al.[13], Häggquist and Söderholm[14], and Leslie et al.[15] pointed out that although the descriptions of uses of remote-sensing data have been published in the literature for decades, research on connecting these uses to improve decision-making and societal outcomes has lagged far behind. Our research aims to fill this important gap by providing a systematic examination of the "teleconnected" cross-hemispheric interactions.

In this study, we quantify the market stabilization benefits of the cross-hemispheric response mechanism triggered by remote-sensing-forecasts using a procedure as presented in Fig. 1. We do this quantification for the following three cases: First and second, a better-than-normal wheat harvest in 2008 and a very poor wheat harvest in 2012 in Russia and Ukraine, and third, a very poor soybean harvest in 2012 in southern Brazil. Our remote-sensing-based forecasting information in Russia and Ukraine captures the forthcoming harvest results for both countries to a satisfactory degree in May, which is two months earlier than the typical wheat harvest season[8–11]. Our remote-sensing-based soybean forecasting information in southern Brazil is derived at the end of April, which is ahead of the planting dates for major soybean producing regions in the Northern Hemisphere (Table 1)[16,17]. We integrated the remote-sensing-based forecasting information with a computable general equilibrium modeling (CGE) approach using the Global Trade Analysis Project (GTAP) database with a land allocation module in two steps (see Methods and Supplementary Table 4 in Supplementary Note 2 for more details). First, we assess the effects of real wheat yield shocks in Russia and Ukraine and soybean production shocks in southern Brazil using historical records from the Food and Agriculture Organization of the United Nations (FAO) and Companhia Nacional de Abastecimento. Then, we establish the supply response scenarios with remote-sensing-based forecasting information, and compared with the real shocks to measure the impact of remote-sensing-based forecasting information on stabilizing the global and regional wheat and soybean markets. Price volatility and household consumption in importing countries are considerably improved by remote-sensing-based forecasting information.

## Results

**Remote sensing forecasting of the wheat and soybean harvests in 2008 and 2012.** The remote-sensing forecasting of wheat in May and June 2008 indicated the production gained by 48.0% and 45.0% over the production level of 2007; and that production in May and June 2012 declined by 15.3% and 19.9% from the production level of 2011 (Table 2). The remote-sensing forecasting of soybean in April 2012 showed a decline of production

**Table 1 Crop calendars for major wheat and soybean producers.**

| | Calendar month for winter wheat | | | | | | | | | | | |
|---|---|---|---|---|---|---|---|---|---|---|---|---|
| | J | F | M | A | M | J | J | A | S | O | N | D |
| **Northern Hemisphere** | | | | | | | | | | | | |
| Canada | | | | | | | H | H | P | P | | |
| E.U. | | | | | H | H | H | P | P | P | P | |
| Russia | | | | | | H | H | P | P | | | |
| Ukraine | | | | | | H | H | P | P | | | |
| USA | | | | | H | H | | | P | P | | |
| **Southern Hemisphere** | | | | | | | | | | | | |
| Argentina | H | | | | P | P | P | P | | | H | H |
| Australia | H | H | | P | P | P | P | | | H | H | H |
| Paraguay | | | P | P | P | | | | H | H | H | |
| South Africa | | | | P | P | | | | | H | H | |
| Uruguay | H | | | P | P | P | P | | | H | H | |
| | **Calendar Month for Soybean** | | | | | | | | | | | |
| | J | F | M | A | M | J | J | A | S | O | N | D |
| **Northern Hemisphere** | | | | | | | | | | | | |
| Canada | | | | P | P | | | H | H | | | |
| E.U. | | | P | P | | | | H | H | | | |
| USA | | | | P | P | | | H | H | | | |
| **Southern Hemisphere** | | | | | | | | | | | | |
| Argentina | P | | | H | H | H | | | | P | P | P |
| Australia | | H | H | H | | | | | | P | P | P |
| Brazil | H | H | H | H | H | | | | P | P | P | P |

Note: **P**: Planting, **H**: Harvesting. While Canada, EU, Russia, Ukraine, and USA produce both winter and spring wheat, this table presents the crop calendars for winter wheat only.
Data source: Agricultural Market Information System, FAO GIEWS[21] Country Brief.

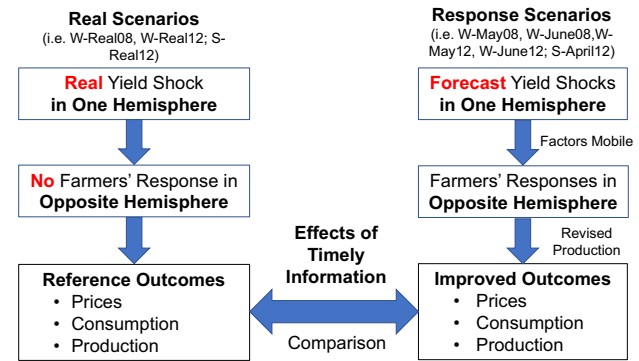

**Fig. 1 Flowchart of the estimation strategy.** Summary information of all scenarios are provided in Supplementary Table 23.

by 18% from the level of 2011 (Table 2). Compared to the real records of FAO[18], the accuracies of our remote-sensing forecasting for wheat were 78% and 74% in May and June 2008, and 56% and 73% in May and June 2012, which are more accurate than those of the World Agricultural Supply and Demand Estimates done by the United States Department of Agriculture (Supplementary Table 1 in Supplementary Note 1). For soybean in 2012, according to Brazil's official statistics[19], virtually all loss of 12% at the national level (10 million tons, in comparison with 2011) was caused by the loss of 35% in southern Brazil (the Brazilian States of Parana, Santa Catarina and Rio Grande do Sul). Our remote-sensing forecasting reported a loss of 44% in southern Brazil. We incorporate these forecasting values into the total factor productivity parameters in the production function of the wheat and soybean sectors in the corresponding regions, respectively.

**Effect on local prices, ceteris paribus**. Abnormal wheat harvests in Russia and Ukraine in 2008 and 2012 provoked changes in real local wheat prices[20] and household wheat consumption, ceteris paribus, in the major importing regions (Table 3). Results suggested that wheat prices decreased by 27–35% under the scenario "W-Real08" and increased by 12–31% in "W-Real12" (Table 3).

The fluctuation extents of country-level prices were reduced by 1.1 to 12.5 percentage points. Taking Japan, one of the largest wheat importers in the world, as an example, the changes in domestic wheat price were −29% under "W-Real08" and +24% under "W-Real12", which were modified to −21% under "W-May08" and +15% under "W-May12", respectively. These ceteris paribus experiments demonstrate that the early availability of reliable wheat harvest information could lead to improved price stabilization on local wheat markets of major importing countries through farmers' reactions in the Southern Hemisphere. The results further show that this cross-hemispheric interaction mechanism could also reduce the fluctuation extents of household wheat consumption by 0.3–2.5 percentage points in 2008 and 0.8–2.0 percentage points in 2012 in the major importing countries (see the last two columns in Table 3). The relatively limited effects on household consumption are attributed to the low price-elasticity of demand for foods.

Similarly, the poor soybean harvest in southern Brazil in 2012 also provoked changes in real local soybean prices and household soybean consumption, ceteris paribus, in the major importing regions (Table 4). The historical shock to soybean production in southern Brazil caused domestic prices of the major importing regions to rise by 12.2–15.9% under the scenario "S-Real12". The

---

**Table 2 Production variations in the three cases in 2008 and 2012 (%).**

| Wheat in Russia & Ukraine | 2008 | 2012 | Soybean in Brazil (SUL) | 2012 |
|---|---|---|---|---|
| Real-shock (Historical records) | 61.2% | −27.2% | Real-shock | −12% (− 35%) |
| May Remote Sensing forecasting | 48.0% | −15.3% | April remote sensing forecasting | −18% (− 44%) |
| June remote sensing forecasting | 45.0% | −19.9% | | |

Note: The changes were relative to the production level in the previous year. For soybean, according to Brazil's official statistics, in the 2012 harvest season, virtually all loss of 12% at the national level (10 million tons, in comparison with 2011) was caused by the loss of 35% in Rio Grande do Sul. The cause was severe drought in the critical growth season.
*Source:* Historical records of wheat production are from FAO[18]. Official statistics of soybean production at the sub-national level in Brazil from Companhia Nacional de Abastecimento is available at https://www.conab.gov.br/info-agro/safras/serie-historica-das-safras?start=30.

---

**Table 3 Impacts of the hemisphere-wise response on local wheat prices and household wheat consumption for 2008 and 2012.**

| | Change in real wheat price [%] | | | Effect of response [percentage point] | | Change in household wheat consumption [%] | | | Effect of response [percentage point] | |
|---|---|---|---|---|---|---|---|---|---|---|
| | W-Real08 (a) | W-May08 (b) | W-June08 (c) | W-May08 (b)–(a) | W-June08 (c)–(a) | W-Real08 (d) | W-May08 (e) | W-June08 (f) | W-May08 (e)–(d) | W-June08 (f)–(d) |
| Bangladesh | −35.2 | −31.8 | −32.0 | 3.4 | 3.3 | 9.1 | 8.0 | 8.0 | −1.1 | −1.1 |
| China | −27.4 | −19.5 | −19.9 | 7.8 | 7.4 | 6.6 | 4.4 | 4.5 | −2.2 | −2.1 |
| Egypt | −35.2 | −33.2 | −33.3 | 2.1 | 2.0 | 9.8 | 9.1 | 9.1 | −0.7 | −0.7 |
| India | −34.0 | −32.0 | −32.1 | 2.0 | 1.9 | 8.8 | 8.1 | 8.1 | −0.7 | −0.6 |
| Japan | −28.5 | −20.6 | −21.0 | 7.9 | 7.5 | 6.9 | 4.7 | 4.8 | −2.2 | −2.1 |
| Korea | −27.6 | −18.4 | −18.8 | 9.3 | 8.8 | 6.7 | 4.1 | 4.2 | −2.5 | −2.4 |
| Nigeria | −27.6 | −21.3 | −21.7 | 6.3 | 5.9 | 6.7 | 5.0 | 5.1 | −1.8 | −1.7 |
| Turkey | −30.2 | −29.1 | −29.1 | 1.2 | 1.1 | 7.7 | 7.3 | 7.3 | −0.4 | −0.3 |
| Middle East | −31.9 | −28.8 | −28.9 | 3.1 | 3.0 | 8.2 | 7.2 | 7.2 | −1.0 | −0.9 |
| Africa | −27.8 | −23.7 | −23.9 | 4.1 | 3.9 | 7.0 | 5.8 | 5.8 | −1.2 | −1.1 |
| | W-Real12 (g) | W-May12 (h) | W-June12 (i) | W-May12 (h)–(g) | W-June12 (i)–(g) | W-Real12 (j) | W-May12 (k) | W-June12 (l) | W-May12 (k)–(j) | W-June12 (l)–(j) |
| Bangladesh | 23.1 | 14.7 | 12.1 | −8.5 | −11.0 | −4.1 | −2.7 | −2.3 | 1.4 | 1.8 |
| China | 19.2 | 11.6 | 9.2 | −7.7 | −10.0 | −3.5 | −2.2 | −1.7 | 1.3 | 1.7 |
| Egypt | 31.0 | 24.7 | 22.7 | −6.3 | −8.2 | −5.7 | −4.7 | −4.4 | 1.0 | 1.3 |
| India | 11.6 | 7.4 | 6.1 | −4.2 | −5.5 | −2.2 | −1.4 | −1.2 | 0.8 | 1.0 |
| Japan | 24.1 | 15.4 | 12.7 | −8.8 | −11.4 | −4.2 | −2.8 | −2.4 | 1.4 | 1.9 |
| Korea | 24.1 | 14.5 | 11.6 | −9.6 | −12.5 | −4.2 | −2.7 | −2.2 | 1.5 | 2.0 |
| Nigeria | 22.6 | 15.4 | 13.2 | −7.2 | −9.4 | −4.0 | −2.9 | −2.5 | 1.2 | 1.6 |
| Turkey | 31.1 | 26.1 | 24.5 | −5.0 | −6.5 | −5.5 | −4.7 | −4.5 | 0.8 | 1.0 |
| Middle East | 24.6 | 17.5 | 15.4 | −7.0 | −9.2 | −4.5 | −3.3 | −2.9 | 1.2 | 1.5 |
| Africa | 23.4 | 16.9 | 15.0 | −6.4 | −8.4 | −4.3 | −3.2 | −2.9 | 1.1 | 1.4 |

Note: Price and consumption changes in W-Real08 and W-Real12 are taken from the FAO's GIEWS database[21] and FAOSTAT database[19].

extent of soybean price rising can be mitigated to a range of −7.3% and −22.3% under the scenario "S-April12". The price rising in Mexico, Japan and Indonesia can be reversed to a moderate decline by −6.4%, −4.5%, and −3.2% owing to the responses in the Northern Hemisphere, mainly because as much as 95%, 80%, and 89% of soybean imports of these three nations were from the United States in 2011, respectively, according to the UN Comtrade[21]. Our results further indicated that while household soybean consumption of these major importers could decrease by −2.3% to −2.9% under the "S-Real12" scenario, under the "S-April12" scenario, the variation range would become between −1.6% and 2.1%, implying a mitigation effect of 1.3–5.0 percentage points (Table 4).

**Responses of farmers to forecast information**. The timely forecast information of abnormal wheat harvests in the Northern Hemisphere could stimulate producers' response in the Southern Hemisphere (Table 5). For the better-than-normal wheat harvest in Russia and Ukraine in 2008, such responses would lead to an aggregated decrease in wheat production by 2.8 and 2.7 million tons under the scenarios of "W-May08" and "W-June08," equivalent to 2.3% and 2.2% of the global wheat export in 2007. In contrast, forecasting of a poor harvest in the Northern

Hemisphere in 2012 would increase wheat production in the South Hemisphere by 2.6 and 3.5 million tons under the scenario of "W-May12" and "W-June12", equivalent to 1.8% and 2.3% of the world export in 2011. Likewise, our analysis suggested that agricultural producers in the United States and Canada responded to early forecasting information on the poor soybean harvest in southern Brazil in 2012 (Supplementary Table 2 in Supplementary Note 1). Producers in the United States and Canada would extend their soybean production by 5% and 7%, resulting in an aggregate increase of soybean supply by 4.4 million tons, which is equivalent to 4.8% of the global soybean export in 2011.

**Effects on the international market**. The cost, insurance and freight (CIF) price of wheat in major importing countries/regions are also stabilized by remote-sensing-based forecasting information (Table 6). Import price stability is especially crucial for nations that heavily depend on external markets for food supply. By effectively utilizing the timely forecasting information in Russia and Ukraine, agricultural producers in the Southern Hemisphere could adjust their wheat production by a proper margin, and thus mitigate the extent of wheat price fluctuation on international markets, which in turn contributes to the stabilization of wheat prices in importing countries. The shock-

**Table 4 Impacts the hemisphere-wise response on local soybean prices and household soybean consumption for 2012.**

| | Change in real soybean price [%] | | Effect on price [percentage point] | Change in household soybean consumption [%] | | Effect on consumption [percentage point] |
|---|---|---|---|---|---|---|
| | S-Real12 | S-April12 | S-April12 | S-Real12 | S-April12 | S-April12 |
| | (a) | (b) | (b)–(a) | (c) | (d) | (d)–(c) |
| China | 15.4 | 2.4 | −13.0 | −2.9 | 2.1 | 5.0 |
| Taiwan (customs territory) | 15.3 | 1.7 | −13.6 | −2.8 | −0.4 | 2.5 |
| Germany | 14.6 | 2.1 | −12.6 | −2.7 | −0.4 | 2.3 |
| Indonesia | 13.8 | −3.7 | −17.5 | −2.6 | 0.7 | 3.3 |
| Japan | 14.9 | −4.5 | −19.3 | −2.7 | 0.9 | 3.7 |
| Mexico | 15.9 | −6.4 | −22.3 | −2.9 | 1.3 | 4.2 |
| Netherland | 15.0 | 2.8 | −12.2 | −2.8 | −0.6 | 2.2 |
| Spain | 15.5 | 8.3 | −7.3 | −2.9 | −1.6 | 1.3 |
| Thailand | 15.2 | 7.0 | −8.2 | −2.8 | −1.4 | 1.5 |
| Africa | 12.2 | −0.4 | −12.6 | −2.3 | 0.1 | 2.4 |
| Asia | 12.8 | 1.0 | −11.9 | −2.4 | −0.2 | 2.2 |

Note: Price and consumption changes in S-Real12 are taken from the FAO's GIEWS database[21] and FAOSTAT database[19].

**Table 5 Responses of wheat farmers in the Southern Hemisphere.**

| | Change in production [%] | | | | | | Change in production [ton] | |
|---|---|---|---|---|---|---|---|---|
| | W-May08 | | | W-June08 | | | W-May08 | W-June08 |
| | Wheat | Grain | Rice | Wheat | Grain | Rice | Wheat | Wheat |
| Argentina | −11.76 | 0.23 | 0.25 | −11.23 | 0.22 | 0.24 | −1,724,255 | −1,646,146 |
| Australia | −7.01 | 0.08 | 0.26 | −6.71 | 0.08 | 0.25 | −758,184 | −725,989 |
| Paraguay | −8.00 | 0.11 | −0.03 | −7.65 | 0.10 | −0.03 | −64,004 | −61,178 |
| South Africa | −11.41 | 0.17 | 0.31 | −10.85 | 0.16 | 0.29 | −217,375 | −206,613 |
| Uruguay | −10.56 | 0.19 | 0.15 | −10.14 | 0.18 | 0.14 | −64,517 | −61,994 |
| Total | | | | | | | −2,828,335 | −2,701,921 |
| | W-May12 | | | W-June12 | | | W-May12 | W-June12 |
| | Wheat | Grain | Rice | Wheat | Grain | Rice | Wheat | Wheat |
| Argentina | 3.31 | −0.06 | −0.07 | 4.43 | −0.08 | −0.09 | 531,309 | 711,057 |
| Australia | 6.39 | −0.03 | −0.09 | 8.49 | −0.04 | −0.13 | 1,752,537 | 2,327,708 |
| Paraguay | 8.15 | −0.11 | −0.07 | 10.99 | −0.15 | −0.10 | 119,258 | 160,928 |
| South Africa | 5.72 | −0.05 | −0.37 | 7.63 | −0.07 | −0.49 | 114,751 | 153,047 |
| Uruguay | 6.77 | −0.10 | −0.08 | 9.13 | −0.14 | −0.11 | 88,117 | 118,704 |
| Total | | | | | | | 2,605,972 | 3,471,443 |

**Table 6 Changes in CIF prices of wheat by major importers for 2008 and 2012.**

| | Change in real wheat price [%] | | | Effect of hemisphere-wise response [percentage point] | |
|---|---|---|---|---|---|
| | W-Real08 (a) | W-May08 (b) | W-June08 (c) | W-May08 (b)–(a) | W-June08 (c)–(a) |
| Bangladesh | −35.4 | −32.0 | −32.1 | 3.4 | 3.3 |
| China | −28.9 | −22.8 | −23.1 | 6.1 | 5.8 |
| Egypt | −36.4 | −34.3 | −34.5 | 2.1 | 2.0 |
| India | −37.3 | −35.2 | −35.3 | 2.1 | 2.0 |
| Japan | −28.5 | −20.6 | −21.0 | 7.9 | 7.5 |
| Korea | −27.7 | −18.4 | −18.9 | 9.3 | 8.8 |
| Nigeria | −27.8 | −21.5 | −21.8 | 6.3 | 6.0 |
| Turkey | −34.8 | −33.6 | −33.7 | 1.2 | 1.2 |
| Middle East | −33.2 | −30.0 | −30.2 | 3.2 | 3.0 |
| Africa | −28.8 | −24.7 | −24.9 | 4.2 | 4.0 |
| | W-Real12 (d) | W-May12 (e) | W-June12 (f) | W-May12 (e)–(d) | W-June12 (f)–(d) |
| Bangladesh | 23.3 | 14.8 | 12.2 | −8.5 | −11.1 |
| China | 22.6 | 13.4 | 10.6 | −9.2 | −12.0 |
| Egypt | 32.8 | 26.1 | 24.1 | −6.7 | −8.8 |
| India | 19.8 | 13.9 | 12.0 | −5.9 | −7.7 |
| Japan | 24.2 | 15.4 | 12.7 | −8.8 | −11.5 |
| Korea | 24.1 | 14.5 | 11.6 | −9.6 | −12.6 |
| Nigeria | 22.9 | 15.6 | 13.4 | −7.3 | −9.5 |
| Turkey | 34.1 | 28.6 | 26.9 | −5.5 | −7.2 |
| Middle East | 26.7 | 19.0 | 16.6 | −7.7 | −10.0 |
| Africa | 24.3 | 17.6 | 15.6 | −6.7 | −8.8 |

smoothing effects of "W-May08" or "W-June08", compared to "W-Real08", ranged between 1.2 and 9.3 percentage points (upper panel of Table 6). Similarly, the shock-smoothing effects of "W-May12" and "W-June12", compared to "W-Real12", would alleviate the price spike on both the global and importer's local markets by 5.5-12.6 percentage points (the lower panel of Table 6). Likewise, for soybean, the shock-smoothing effects of "S-April12", compared to "S-Real12", would reduce soybean price inflation to a range of −6.4% (in Mexico) and 8.3% (in Spain) (Supplementary Table 3 Supplementary Note 1).

**Robustness of the results**. A major concern about the robustness of CGE simulation results lies in parameter uncertainty, given the fact that the elasticity parameters adopted in a CGE model are typically taken from the literature of empirical econometric estimations[22,23]. We test the robustness of our primary results against elasticity parameters assumed in this analysis (Supplementary Table 4 in Supplementary Note 2). Our tests are run for ±30% of the parameter values that could influence the results of the major simulations. These parameters include the Armington elasticity for the grain sectors (i.e., wheat, other cereals, and oil crops for the wheat analysis and these sectors plus soybean for the soybean simulations) and value-added substitution elasticity for food-related sectors (the same as the robustness tests for the Armington elasticity), the elasticity of substitution between food products for the household, as well as the elasticity of land transformation. In addition, we run a set of sensitivity tests by halving the substitution elasticities of production factors for food-related sectors. The testing results are qualitatively robust against the three sets of elasticities, although moderate quantitative variations are present. Supplementary Note 2 reports the quantitative variations of local wheat and soybean price changes in importing regions (Supplementary Tables 5–12), the change in household wheat and soybean consumption in importing countries (Supplementary Tables 13–20), and farmers' responses in the

Southern and Northern Hemispheres (Supplementary Tables 21 and 22), respectively.

## Discussion

Significant price volatility of agricultural commodities is a threat to global food security and a challenge for the world's policy makers. Reliable and timely forecasting of food production, a unique application of satellite Earth observations, can stabilize the agricultural market. Our simulations of the world-trade general equilibrium model showed that satellite-based timely, accurate and reliable forecasts of wheat and soybean harvest in major exporting countries could stimulate responses by producers of the opposite Hemisphere. Utilizing the seasonal lags in crop calendars between the two hemispheres, producers of the opposite Hemisphere could adjust their plans to address the foreseen supply-demand gap in world markets. Our results also indicate that such a response could reduce price volatilities in world food and feed markets.

The inter-hemispheric system brings about three major benefits to the world economy. First, the cross-hemisphere response mechanism can help stabilize global agricultural markets. In other words, such a global hedging mechanism lessens the risk premium of consumers or producers of agricultural/food commodities with smaller price fluctuations, assuming that the majority of purchasers or farmers have a risk-averse utility function. Second, this mechanism can enhance the efficiency of resource allocation in the agricultural sectors. One of the conditions to establish a perfectly competitive market is perfect information. The satellite-based harvest prediction information can be used to reduce deadweight losses on the international and indigenous markets, leading to more efficient allocation of labor, capital, land, and other resources. Third, the mechanism has the potential to make speculators such as hedge funders hesitate to invest in agricultural commodities when abundant or poor crops are convincingly reported before harvest. If investors perceive the farmers'

adjustment actions, the expected excessive profits from speculative activities would be diminished.

One particular policy implication is about trade restrictions. The cross-hemispheric system works based on price transmission between international and local markets. Any disturbance of international market synchronization such as export and import restrictions wanes the functionality of the inter-hemispheric response mechanism. For instance, if export restrictions are imposed by Australia, the spillover effect induced by the land allocation adjustment of Australian farmers would not be fully conveyed to the world market. For a similar reason, lowering import trade barriers by food-deficit countries facilitates the receiving of the benefit from the inter-hemispheric responses. Although historically, national governmental bodies occasionally regulated the export or import of agricultural goods, countries that imposed export restrictions could receive a substantial welfare loss mainly because farmers in exporting countries lost the opportunity to sell their crop products to foreign buyers, leading to excess supply and lower prices in the domestic market as shown by Tanaka and Hosoe[23,24]. Besides, an export restriction can cause a backfire of speculation on the international market and lead to a rise in import prices of beef and pork in the following year. To evaluate this second-stage impact in a CGE setting is a topic for future research.

Another policy implication is to support the research and development of timely and more accurate crop harvest forecast and improve the accessibility to such forecasting information. We uncovered the benefits of the forecasting information which was one or two months earlier than the conventional harvest information. However, the accuracies of our forecasting for wheat in Russia and Ukraine were only 78% and 74% in May and June 2008, and 56% and 73% in May and June 2012. If the forecasting accuracy can be significantly improved to over 90% in May, meaning two months before harvest, more farmers in the Southern Hemisphere could respond to the information with fuller confidence and longer time window, which would significantly enhance the functionality of the inter-hemispheric supply response mechanism in improving the global food security.

## Methods

**Choice of the three cases.** It is worth noting that a valid case for the general equilibrium simulations needs to simultaneously meet the following three conditions: (1) the availability of a global social accounting matrix (SAM) in the year ($t$–1) prior to the year ($t$) with a good or poor harvest, (2) the best available remote-sensing forecasting which is able to capture the good or poor harvest in the year ($t$), (3) the sufficiently large supply variation caused by the good or poor harvest which is able to exert a noticeable impact on the global food market. We have done this match across all published SAMs by the GTAP-v10 database and the best available remote-sensing forecasting results of wheat, corn, and soybean harvests in the Global Agriculture Monitoring System of NASA Harvest Program (https://glam.nasaharvest.org/). Our choice of the soybean case is further enabled by the recently published remote-sensing dataset[17]. As a result, we identified two cases for wheat, and one case for soybean, but did not find a valid case for corn.

**Remote sensing estimates of production.** Pre-harvest assessment of crop yield has been an important research theme since the 1970s[15,25]. "The Large Area Crop Inventory Experiment" project launched in 1974 by the United States Department of Agriculture, the National Aeronautics and Space Administration, and the National Oceanic and Atmospheric Administration demonstrated that crop monitoring from space could supply essential pre-harvest information on production in terms of precision and timeliness[26]. A variety of methods have been developed to estimate crop yields using remotely sensed information. One stream of the research employs biophysical crop-simulation models to retrieve crop growth parameters from remotely sensed data and then, calibrates and drives the models based on these parameters. Such crop-simulation models include CERES[27], WOFOST[28], CROPSYST[29], and STICS[30]. However, these models typically require numerous crop-specific inputs such as soil characteristics, management practices, agro-meteorological data, and phonological dates to simulate crop growth and development through the crop cycle[31,32]. In contrast, the statistical regression-based approaches are typically more straightforward to implement and do not require numerous inputs because they work with the empirical relationships between historical records of yields and reflectance-based vegetation indices and agrometeorological data[15,25]. Within this second stream of research, Fischer[33] showed the possibility to forecast wheat yields using leaf area at the onset of the reproductive stage. Tucker et al. [34] found significant linear correlations between wheat yields and time-integrated Normalized Difference Vegetation Index (NDVI) values during the growing period. They further demonstrated that the strongest correlation between yields and NDVI appeared around the time of maximum green leaf biomass. Pinter et al. [35] showed strong correlations between wheat yields and accumulated NDVI during the growing season. Based on the accumulative progress in this stream of research, numerous studies have related spectral vegetation indices to crop yields in a variety of regions and countries[36–38].

Our wheat yield forecasting for Russia and Ukraine is based on the method presented in Becker-Reshef et al. [8] and its latest development[9–11]. Becker-Reshef et al. [8] developed a generalized empirical model for forecasting winter wheat yield and production using remote sensing data and official statistics. The model establishes a robust relationship between the yields, the seasonal peak NDVI derived from Moderate Resolution Imaging Spectro-radiometer and maximum winter wheat percentage per corresponding administrative units of the country. The word "generalized" here means that the relationship between the maximum NDVI signal of pure winter wheat pixels and yield established by the two-step procedure of the method is transferable and directly applicable at the state and national levels. The method was further improved by Franch et al. [10] to incorporate Growing Degree Day information into the model to improve the timeliness of the forecasts. The method has been applied for multiple countries including the USA, Ukraine and China, and these applications have produced accurate forecasts of wheat production at the national/state level 1-2 months before harvest. In this research, the timing of remote sensing monitoring was on 10th of May and 9th of June in 2008 and 2012 for Russia and on 5th of May and 4th of June in 2008 and 2012 for Ukraine, respectively. Given the frequency of remotely sensed data, such models can provide forecasts that are updated on a daily or weekly basis. Supplementary Figure 1 in Supplementary Note 3 presents the positive NDVI anomaly in 2008 and the negative NDVI anomaly in 2012 during the critical wheat growing season in Russia and Ukraine.

Our soybean forecasting information is derived from the data and methods reported in Song et al. [16,17]. Song et al. [17] established a machine-learning-based turnkey classification model to characterize soybean cover across the South American continent at 30 m spatial resolution and in the soybean growth season every year. The model was constructed based on three years (2017 through 2019) of continentally distributed field

observation as training and consistently processed Landsat and Moderate Resolution Imaging Spectroradiometer satellite observations as input. With operational satellite acquisitions, the established model can be applied in a back-casting mode to map historical soybean cultivation as well as in a forecasting mode to map future soybean cultivation. Although the direct output of the model is a soybean classification map, crop areas derived from the high-resolution map is a close surrogate to soybean production for the following reason. To be mapped as soybean, a pixel must present a complete growth cycle across a growing season as well as sufficient greenness in the spectral feature space. Therefore, the mapped soybean pixels represent those actively cultivated fields with harvestable yield. Failed crops resulting from weather anomalies that show an incomplete growth cycle or reduced greenness are not mapped. In this research, the remote-sensing-based soybean production anomaly in 2012 in southern Brazil (the Brazilian States of Parana, Santa Catarina and Rio Grande do Sul) represented the information available on April 30th, 2012 (Supplementary Figure 2 in Supplementary Note 3).

**The establishment of scenarios and the estimation strategy.** We set up the corresponding CGE model for each of the three cases to demonstrate the effects of timely forecasting information on wheat and soybean production in Russia-Ukraine and southern Brazil. In the three CGE models, we select 2007 and 2011 as the base years (i.e., the base-year scenarios "W-Base07", "W-Base11", and "S-Base11"). This selection is justified by the following two reasons. First, the GTAP-v10 database[39] provides the balanced social accounting matrices (SAM) for the world economy in 2007 and 2011, which are indispensable for the construction of the three CGE models. Second, the focus of this research is on the potential market stabilizing effect of the inter-hemispheric supply response mechanism triggered by a timely forecast of crop harvests. A very good wheat harvest in 2008 compared to 2007 and a very bad wheat harvest in 2012 compared to 2011 in Russia and Ukraine, and an extremely poor soybean harvest in 2012 in Rio Grande do Sul in Brazil compared to 2011, in combination with the availability of the SAMs for the world economy in 2007 and 2011, provide a unique opportunity for realizing the mission of this research (cf. "*Choice of the three cases*" above).

The real level of wheat production in 2008 in the regions (i.e., the Russia-Ukraine region) increased by 61.2%, and that in 2012 dropped by 27.2%, respectively, in comparison with the level in the previous year (Table 2). We incorporate such extent of production increase and decrease into the total factor productivity parameters in the production function of the wheat sector in the Russia-Ukraine region (the scenarios "W-Real08" and "W-Real12") and that of the soybean sector in Brazil (S-Real12). In the real-shock scenarios, we do not allow factor mobility across sectors and regions in all the countries except Russia-Ukraine or Brazil in each model because the news on the harvest variations in these three countries came in the harvest period and was too late for any proactive adaptation in the counterpart Hemisphere regions (the left panel in Fig. 1).

In the cross-hemispheric response scenarios "W-May08," "W-June08," "W-May12," and "W-June12," we allow rational responses of producers in the Southern Hemisphere to reduce or augment wheat planting when they receive the prompt forecasting information on the forthcoming harvests in the Russia-Ukraine region, which were an increase of 48% and 45% in the May and June forecasting in 2008 and a decrease of 15.3% and 19.9% in the May and June forecasting in 2012 (Table 2). In the same way, we assume that rational farmers in the Northern Hemisphere response to early prediction data of soybean production in Brazil, which is an 18% decline in production (Table 2). The extent of the response

in re-allocating production factors is endogenously determined by the market opportunity that emerged as a result of the anticipated harvests. This means that we allow factor mobility for the response scenarios "W-May08," "W-June08," "W-May12," "W-June12" and "S-April12" being subject to the constraint of the wheat and soybean crop calendars as presented in Table 1. In more detail, under the response scenarios, wheat farmers in all four Southern Hemisphere countries (Argentina, Australia, Paraguay and South Africa) can make crop planting decisions in response to price variations induced by the early forecasting information in May or June in 2008 or 2012. By the same token, soybean farmers in the Northern Hemisphere (i.e., the U.S. and Canada) can adjust their soybean planting areas according to the remote-sensing information in April (the right panel in Fig. 1). The technical procedure of implementing the cross-hemispheric response scenarios includes two steps (the right panel in Supplementary Figure 3, Supplementary Note 3). In the first step, the production shocks forecasted in May, June, or April drive the allocation of production factors in the Southern or Northern Hemisphere in an endogenous manner. This step produces intermediate equilibrium. In the second step, the real-shocks of harvest failures in the Russia-Ukraine region or Brazil that occurred in 2008 or 2012 are introduced to the above-mentioned intermediate equilibrium setting to produce final equilibriums. In this step, the allocation for the factors of production is fixed or immobile across sectors. Supplementary Table 23 in Supplementary Note 3 summarizes the scenarios explained above, together with the information on factor mobility.

**World trade CGE models.** The general equilibrium theory proposed by L. Walras was refined by K. Arrow and G. Debreu, two Nobel Prize Laureates of Economics, to analyze the existence and stability of competitive equilibrium. Computable general equilibrium (CGE) models were originated from input-output models pioneered by Wassily W. Leontif and architected initially in Johansen[40] and Harberger[41]. A CGE model is formulated as a nonlinear programming problem and is built on an integrated system of equations whose simultaneous solution determines values of endogenous variables. The underlying equations in the system are derived from economic theory to represent the behavior of economic agents and markets. The model is capable of conducting commodity and price analysis with detailed national accounts and international trade data. For example, in comparison with the multi-regional input-output modeling in explaining land-use change[42–44], the advantage of CGE is that it can model the endogenous substitution across alternative production choices and land-use types, which is driven by cost-benefit calculations of economic agents under the condition of market competition. In this way, it avoids the limitation of fixed land-use coefficients attached to individual sectors in the multi-regional input-output modeling.

We construct three world-trade computable general equilibrium (CGE) models with a land allocation module based on the GTAP-v10 database for SAMs in 2007 and 2011[39]. These SAMs encompass 141 customs territories/regions and 65 industrial sectors. We aggregated them into 23 regions and 13 sectors (Supplementary Table 24 in Supplementary Note 3) for the wheat sector analysis and 19 regions and 14 sectors for the soybean sector analysis (Supplementary Table 25 in Supplementary Note 3, and Supplementary Note 4). In this way, we can focus on large wheat or soybean importers and exporters for the regional aggregation and on the agricultural and food-related commodities/sectors for the sectoral aggregation. For readers who are not familiarized with CGE models, please refer to Supplementary Note 5 in which we present general information on CGE and also detailed specifications of the model. Our CGE model follows the

principles of the standard CGE models developed by Devarajan et al. [45] (Supplementary Figures 4). We extend the standard model to the global scale with three global SAMs from the GTAP database and introduce several modifications to enhance the representation of behaviors of different agents and markets.

It is assumed that each sector maximizes its profit with the Leontief technology and that a constant elasticity of substitution (CES) function aggregates the factors of production such as labor, capital, and farmland, with elasticities from the GTAP database and recent literature. Using the constant elasticity of transformation form, the domestically produced goods are distributed between a composite export and domestic good, which is combined with an import aggregate good by the Armington function[46]. The Armington elasticities from the GTAP database are for the medium to long-run analysis. To gauge the short-term impacts of farmers' response to remote sensing forecasting information, those elasticities for food-related sectors are halved following the estimations of Bajzik et al. [47].

Our CGE model allows for substitution across food-related goods in household consumption with the constant elasticity of substitution (CES) form (Supplementary Figure 5). The value of substitution elasticity is established based on the estimations of price elasticity of demand for cereal goods[48]. The relationship between the price elasticity of demand and the elasticity of substitution is given in Shoven and Whalley[49].

We replace the perfect farmland mobility assumption between sectors in the standard model with three-level constant elasticity of technology (CET) functions similar to those employed in conventional land-use CGE models[50]. At the first level, the land is assigned for livestock and crop aggregate (Supplementary Fig. 6). The second level specifies the substitutive relationship between cereal-oil crop composite and other crops. The third level specifies the substitutive relationship across wheat, coarse grain, and oilseeds (including soybean in the soybean analysis). With the land allocation specification, the model is allowed to describe the optimized farmers' planting decision-making. The elasticities of substitution between livestock and aggregated crop, between two crop groups, are assumed to be $-0.2$ and $-0.5$, respectively, following the existing literature[50,51]. Haile et al. [51] estimated the short-run growing-area elasticities with respect to own crop prices for each major global producer of wheat and soybean, which are suitable for the purpose of this research.

**Limitations**. Several limitations of our research are worth mentioning. First, we assume that agricultural producers in the Southern and Northern Hemispheres can choose a more profitable crop mixture in response to the timely harvest forecasting. In practice, however, some of them may not be able to react to such information owing to the constraints of crop rotation and other pre-existing conditions in their farmland. Ignorance of such constraints would lead to an overestimation of the benefits of the cross-hemispheric response mechanism.

Second, in our simulations, we focus on wheat or soybean price variations which are exclusively associated with the very good and bad harvests in the Russia-Ukraine region and southern Brazil. Yet, in reality, speculative activities may have further destabilized food markets with additional long and short positions immediately after the good and bad harvests were predicted and reported to the public. If a huge amount of speculative money was invested in wheat or soybean commodity as soon as it was announced and farmers in the Southern or Northern Hemisphere reckoned the effect in their expectation, the cross-hemispheric reactions and the beneficial effect would become greater. Therefore, our analysis could underestimate the beneficial effect of the cross-hemispheric response mechanism.

Third, we assume in the simulations for simplicity that only five countries in the Southern Hemisphere for the wheat sector analysis and two countries in the Northern Hemisphere for the soybean sector analysis utilize the pre-harvest information to adjust the land allocation for crops. If more farmers in the opposite Hemisphere regions responded to early information, the international market adjustment would have been larger, meaning that our research could again underestimate the benefits of the interhemispheric effects.

The fourth limitation concerns the annual timeframe of our CGE model. It assumes that production in one hemisphere is exogenous, and only farmers in the opposite hemisphere can adjust their production. This approach focuses on a particular trajectory in a recursive semi-annual setting, as recommended by Gouel[52] and Miranda & Glauber[53]. While this configuration effectively demonstrates the economic logic underlying the cross-hemispheric response mechanism, extending our model to a recursive semi-annual framework could offer more insights. Specifically, it could enhance our understanding of how precise harvest forecasting information impacts intra-annual price volatility.

## Data availability

The MODIS Surface Reflectance 8-Day 250 m data is distributed by NASA EOSDIS Land Processes Distributed Active Archive Center, https://doi.org/10.5067/MODIS/MOD09Q1.061 and can be accessed from: https://lpdaac.usgs.gov/products/mod09q1v061/. The annual soybean maps over Brazil can be viewed and downloaded at https://glad.earthengine.app/view/south-america-soybean. The data for the SAMs are taken from the GTAP database version 10 (https://www.gtap.agecon.purdue.edu/databases/v10/index.aspx). The historical data of wheat production are from FAOSTAT (https://www.fao.org/faostat/en/). Official statistics of soybean production at the subnational level in Brazil are available at https://www.conab.gov.br/info-agro/safras/serie-historica-das-safras?start=30.

## Code availability

The general algebraic modeling system codes used for this analysis are available upon reasonable request.

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

## Acknowledgements

This research was supported by NASA Applied Sciences Cooperative Agreement #80NSSC18M0039 (NASA Harvest Phase 1), NASA LCLUC Program (grant number: 80NSSC20K1490), and the Japan Society of Promotion of Science (JSPS) KAKENHI Grant (grant number: 23H02317).

## Author contributions

T.T. conceived the original idea, and T.T. and L.S. designed the research. I.B. and X.S. generated the remote sensing crop forecast data. T.T. developed the models and conducted the simulations. T.T., L.S., and X.S. wrote the manuscript. T.T., L.S., I.B., X.S., and E.P. commented on the manuscript and discussed the results at all stages.

## Competing interests

The authors declare no competing interests.
