## [Peer Review File · Communications Earth & Environment]

3rd Apr 23

Dear Professor Sun,

Your manuscript titled "Satellite forecasting of crop harvest can trigger cross-hemispheric production response and improve global food security" has now been seen by 2 reviewers, whose comments are appended below. You will see that they find your work of some potential interest. However, they have raised quite substantial concerns that must be addressed. In light of these comments, we cannot accept the manuscript for publication, but would be interested in considering a revised version that fully addresses these serious concerns.

Specifically, for publication in Communications Earth & Environment to be appropriate, we will need you to:

- Provide compelling new insights that go beyond previous research on farmers' responses to harvest forecasts between the two hemispheres, and discuss the impact of more accurate harvest information on price volatility
- Demonstrate that crop forecasting using Earth observation methods is more accurate and timely than that done by governments/international organizations
- Ensure that your methods and approach are fully described and justified, including the choice of studied crops, regions and scenarios used, and the temporal resolution of your model simulations

We hope you will find the reviewers' comments useful as you decide how to proceed. Should additional work allow you to address these criticisms, we would be happy to look at a substantially revised manuscript. If you choose to take up this option, please either highlight all changes in the manuscript text file, or provide a list of the changes to the manuscript with your responses to the reviewers.

If the revision process takes significantly longer than three months, we will be happy to reconsider your paper at a later date, as long as nothing similar has been accepted for publication at Communications Earth & Environment or published elsewhere in the meantime.

We understand that due to the current global situation, the time required for revision may be longer than usual. We would appreciate it if you could keep us informed about an estimated timescale for resubmission, to facilitate our planning. Of course, if you are unable to estimate, we are happy to accommodate necessary extensions nevertheless.

Please use the following link to submit your revised manuscript, point-by-point response to the reviewers' comments with a list of your changes to the manuscript text (which should be in a

separate document to any cover letter) and any completed checklist:

[link redacted]

Please do not hesitate to contact me if you have any questions or would like to discuss the required revisions further. Thank you for the opportunity to review your work.

Best regards,

Alienor Lavergne, PhD
Associate Editor
Communications Earth & Environment

EDITORIAL POLICIES AND FORMAT

If you decide to resubmit your paper, please ensure that your manuscript complies with our editorial policies and complete and upload the checklist below as a Related Manuscript file type with the revised article:

Editorial Policy Policy requirements (Download the link to your computer as a PDF.)

For your information, you can find some guidance regarding format requirements summarized on the following checklist:(<https://www.nature.com/documents/commsj-phys-style-formatting-checklist-article.pdf>) and formatting guide (<https://www.nature.com/documents/commsj-phys-style-formatting-guide-accept.pdf>).

REVIEWER COMMENTS:

Reviewer #1 (Remarks to the Author):

General comments

The question of how well earth observation data can inform policymakers is of great interest. The authors make a case that improved and more timely information can reduce uncertainty in commodity markets and stabilize markets by signaling to producers in the northern (southern) hemisphere when there are production shocks in the southern (northern) hemisphere, allowing them to compensate by increasing or decreasing plantings accordingly.

There are two features that are missing from the model which would allow the authors to test their hypothesis. First, the model is an annual, yet production is occurring effectively twice within the

year. Recent papers by Gouel (2020) and Miranda and Glauber (2021) show how price expectations at planting are affected by current information. Incorporating this model feature would allow the researchers to directly test the impact of more accurate information about the harvest on price volatility.

Gouel, C. 2020. "The Value of Public Information in Storable Commodity Markets: Application to the Soybean Market." *American Journal of Agricultural Economics*. <https://doi.org/10.1002/ajae.12013>
Miranda, M. and J. Glauber. 2021. "A Model of Asynchronous Bi-Hemispheric Production in Global Agricultural Commodity Markets." *American Journal of Agricultural Economics*
<https://doi.org/10.1111/ajae.12241>

Second, the paper assumes that the EO data is superior to other forecasting methods. Forecasts are typically available from US Department of Agriculture, the International Grains Council, and FAO as early as May for crops harvested later during the year in both the southern and northern hemispheres. Demonstrating that the forecast accuracy of EO methods is more accurate (or more timely) than the forecasts of the USDA, IGC or FAO would provide a compelling rationale for why EO could lead to improved food security. There is a lot of work in the literature on forecast accuracy of USDA reports:

Isengildina-Massa, O., S.H. Irwin, and D.L. Good. 2006. "Are Revisions to USDA Crop Production Forecasts Smoothed?" *American Journal of Agricultural Economics* 88(4): 1091-1104. Accessed at: <https://www.jstor.org/stable/4123548>

Isengildina-Massa, O., S.H. Irwin, D.L. Good, and J.K. Gomez. 2008. "Impact of WASDE Reports on Implied Volatility in Corn and Soybean Markets." *Agribusiness* 24 (4): 473– 490. Accessed at <https://doi.org/10.1002/agr.20174>

Isengildina-Massa, O., B. Karali, and S.H. Irwin. 2013. "When do the USDA forecasters make mistakes?" *Applied Economics* 45:36, 5086-5103. Accessed at: <https://doi.org/10.1080/00036846.2013.818213>

Xiao, J., C. E. Hart, and S. H. Lence. 2017 "USDA Forecasts of Crop Ending Stocks: How Well Have They Performed?", *Applied Economic Perspectives and Policy* 39(2), 220–241. Accessed at: <https://doi.org/10.1093/aep/px023>

Reviewer #2 (Remarks to the Author):

The article is well-structured and tackles a highly relevant subject. The literature review and methodological sections are clear and provide the right level of detail to understand the analyses developed in the following sections. The limitations are clearly highlighted and discussed. Unfortunately, I cannot comment on the quality of the statistical analysis related to the World trade CGE models used in the paper.

Some comments are below.

1) A visual representation of the general modelling strategy (from data sourcing to output) would

help the reader to better approach the complexity of the analysis.

2) Why are you focusing only on "wheat or soybean price variations which are exclusively associated with the very good and bad harvests in the Russia-Ukraine region and southern Brazil"?

3) Why are you not including speculative activities in your simulation?

Response to Reviewer's Comments on Manuscript COMMSENV-23-0072

Title: Satellite forecasting of crop harvest can trigger cross-hemispheric production response and improve global food security

Overall Response: We would like to thank the editor and the two reviewers for their detailed comments and suggestions, which are very helpful to improve the quality of the manuscript. In the revision, we have taken all the comments and suggestions into consideration and addressed all the raised questions and concerns. Changes can be easily seen in the "track changes" version of the revision, which is also submitted. In this letter, we report the revision point by point in response to the comments of the reviewers.

Reviewer 1

The question of how well earth observation data can inform policymakers is of great interest. The authors make a case that improved and more timely information can reduce uncertainty in commodity markets and stabilize markets by signaling to producers in the northern (southern) hemisphere when there are production shocks in the southern (northern) hemisphere, allowing them to compensate by increasing or decreasing plantings accordingly.

There are two features that are missing from the model which would allow the authors to test their hypothesis. First, the model is an annual, yet production is occurring effectively twice within the year. Recent papers by Gouel (2020) and Miranda and Glauber (2021) show how price expectations at planting are affected by current information. Incorporating this model feature would allow the researchers to directly test the impact of more accurate information about the harvest on price volatility.

*Gouel, C. 2020. "The Value of Public Information in Storable Commodity Markets: Application to the Soybean Market." *American Journal of Agricultural Economics*. <https://doi.org/10.1002/ajae.12013>*

*Miranda, M. and J. Glauber. 2021. "A Model of Asynchronous Bi-Hemispheric Production in Global Agricultural Commodity Markets." *American Journal of Agricultural Economics* <https://doi.org/10.1111/ajae.12241>*

RESPONSE: Thanks a lot for confirming the value of our research. With regard to the annual setting of our model, we would like to make the following clarification. In our simulations, we focus on the scenario where farmers in Hemisphere A receive forecast information for harvest in Hemisphere B and then use this information to adjust their planting plan by leveraging the seasonal lags in crop calendars between the two Hemispheres (Table 1). In other words, the production in Hemisphere B is exogenous "because after planting, changes to production forecasts can reasonably be considered exogenous" (Gouel, 2020) and endogenous adjustment in production can be made by farmers in Hemisphere A only. Such a setting takes one specific trajectory in a recursive semi-annual setting as suggested by Gouel (2020) and Miranda & Glauber (2021). The advantage of our setting is that it can clearly reveal and simulate the economic logic underpinning the following cross-hemispheric response mechanism:

If poor wheat harvests are reported by reliable harvest forecast one to two months before the harvest, say, in May, for the Northern Hemisphere, the markets would anticipate a wheat supply shortage and price rise in the coming wheat marketing year. This expectation would incentivize farmers in the Southern Hemisphere to allocate more farmlands for wheat planting, which would alleviate the expected supply shortage and price spike thanks to the additional wheat supply arriving in October-February (Table 1) from the Southern Hemisphere. By contrast, if good wheat harvests are predicted accurately in May for the Northern Hemisphere, farming operators in the Southern Hemisphere would be incentivized to allocate less farmlands for wheat, which would reduce wheat exports from the Southern Hemisphere in the coming wheat marketing year, mitigating the extent of price decline on the global market. Thus, the inter-hemispheric supply response mechanism triggered by timely forecast of crop harvests would work as an agricultural market stabilizer. The same logic is applicable to the case of soybean (Table 2). Southern Brazil's soybean harvest prediction information for April could be utilized for farming decisions in the Northern Hemisphere regions such as the United States and Canada, where 61% of the world's soybean production was produced in 2011 according to the FAOSTAT. Our study shows that such an inter-hemispheric supply response can be captured by an annual CGE model if the shock is sufficiently large.

On the other hand, we have added the following paragraph in the "Limitation" subsection of Methods (page 14, lines 421-428):

"The fourth limitation concerns the annual timeframe of our CGE model. It assumes that production in one hemisphere is exogenous, and only farmers in the opposite hemisphere can adjust their production. This approach focuses on a particular trajectory in a recursive semi-annual setting, as recommended by Gouel⁵³ and Miranda & Glauber⁵⁴. While this configuration effectively demonstrates the economic logic underlying the cross-hemispheric response mechanism, extending our model to a recursive semi-annual framework could offer more insights. Specifically, it could enhance our understanding of how precise harvest forecasting information impacts intra-annual price volatility."

Second, the paper assumes that the EO data is superior to other forecasting methods. Forecasts are typically available from US Department of Agriculture, the International Grains Council, and FAO as early as May for crops harvested later during the year in both the southern and northern hemispheres. Demonstrating that the forecast accuracy of EO methods is more accurate (or more timely) than the forecasts of the USDA, IGC or FAO would provide a compelling rationale for why EO could lead to improved food security. There is a lot of work in the literature on forecast accuracy of USDA reports:

Isengildina-Massa, O., S.H. Irwin, and D.L. Good. 2006. "Are Revisions to USDA Crop Production Forecasts Smoothed?" American Journal of Agricultural Economics 88(4): 1091-1104. Accessed at: <https://www.jstor.org/stable/4123548>

Isengildina-Massa, O., S.H. Irwin, D.L. Good, and J.K. Gomez. 2008. "Impact of WASDE

Reports on Implied Volatility in Corn and Soybean Markets.” Agribusiness 24 (4): 473–490. Accessed at <https://doi.org/10.1002/agr.20174>

Isengildina-Massa, O., B. Karali, and S.H. Irwin. 2013. “When do the USDA forecasters make mistakes?” Applied Economics 45:36, 5086-5103. Accessed at: <https://doi.org/10.1080/00036846.2013.818213>

Xiao, J., C. E. Hart, and S. H. Lence. 2017 “USDA Forecasts of Crop Ending Stocks: How Well Have They Performed?”, Applied Economic Perspectives and Policy 39(2), 220–241. Accessed at: <https://doi.org/10.1093/aep/px023>

RESPONSE: There is a large body of publications evaluating the accuracy of the agricultural market forecasts produced by the USDA and other agencies and further evaluate the market’s reactions to these forecasts. The USDA forecasts are regarded as one of the most informative and influential sources for agricultural production prognosis (Sanginabadi, 2017). **However, there is no attention in the literature on the cross-hemispheric response mechanism between farmers and timely RS-based forecasts.** In the business world, although it is reported that soybean and corn producers in Brazil frequently monitor developments in the US Corn Belt and the US farmers do the same to monitor developments in the Southern Hemisphere (Hubbs, 2018), **a systematic examination of such “tele-connected” interaction is still missing. Our research aims to fill this important gap.** This means that the focus of our research is on this “tele-connected” interaction rather than demonstrating that our RS-based forecasting is better than those of the USDA, IGC or FAO.

Despite of the above, the RS-based estimation techniques developed by Becker-Reshef et al. (2010), Franch et al. (2015, 2019), and Skakun et al. (2017) do produce more accurate forecasting than the USDA-WASDE. We present this comparison in a new table (Table S1 of this version) and cite the table in the main text as follows (page 3, lines 84-88):

“Compared to the real records of FAO18, the accuracies of our remote-sensing forecasting for wheat were 78% and 74% in May and June 2008, and 56% and 73% in May and June 2012, which are more accurate than those of the World Agricultural Supply and Demand Estimates (WASDE) done by the United States Department of Agriculture (USDA) (Table S1).”

We opt not to present this table in the main text of the manuscript so that the manuscript has a clear-cut focus.

Table S1. Forecasting of Wheat Production (million tons) in Russia & Ukraine in 2008 and 2012: A comparison

	2008			2012		
	Real prod.	Our RS forecasting	WASDE forecasting	Real prod	Our RS forecasting	WASDE forecasting
	89.7			53.5		
May		80.1	72.0		66.2	76.0
June		79.5	75.0		62.6	74.0

Note: The changes were relative to the production level in the previous year.

Source: Historical records of wheat production are from FAO (2021). The forecasting data of the USDA-WASDE are from USDA (2023).

References:

FAO, FAOSTAT database (FAO, Rome, Italy, 2021), available at: <http://www.fao.org/faostat/en/#data/QC>

Hubbs, T. Soybean marketing decisions difficult. (AgriView, 2018) Source URL: https://www.agupdate.com/agriview/markets/crop/soybean-marketing-decisions-difficult/article_378658b4-e4e0-58c6-8ca8-230c44a373ab.html.

Sanginabadi, B. USDA Forecasts: A meta-analysis study. Manuscript, Economics Department, University of Hawaii at Manoa (2017).

USDA (2023), The World Agricultural Supply and Demand Estimates (WASDE). Source URL: <https://www.usda.gov/oce/commodity/wasde>.

Reviewer 2

The article is well-structured and tackles a highly relevant subject. The literature review and methodological sections are clear and provide the right level of detail to understand the analyses developed in the following sections. The limitations are clearly highlighted and discussed. Unfortunately, I cannot comment on the quality of the statistical analysis related to the World trade CGE models used in the paper.

RESPONSE: Thank you very much for confirming the contributions of the manuscript.

Some comments are below.

1) A visual representation of the general modelling strategy (from data sourcing to output) would help the reader to better approach the complexity of the analysis.

RESPONSE: Thanks a lot for this suggestion. We have designed two visual representations of the general modelling strategy (Figure 1 for general readership and Figure S3 for CGE specialists), and extended “*The establishment of scenarios*” subsection in Methods to “*The establishment of scenarios and the estimation strategy*” to explain our modeling strategy (pages 10-12, lines 297-344).

Fig 1. Flowchart of the estimation strategy. Summary information of all scenarios are provided in Table S23.

Figure S3: The implementation procedure of the hemisphere-wise response estimation

2) Why are you focusing only on "wheat or soybean price variations which are exclusively associated with the very good and bad harvests in the Russia-Ukraine region and southern Brazil"?

RESPONSE: Please note that a valid case for the general equilibrium simulations needs to simultaneously meet the following three conditions: (1) the availability of a global social accounting matrix (SAM) in the year ($t - 1$) prior to the year (t) with a good or poor harvest, (2) the best available RS forecasting which is able to capture the good or poor harvest in the year (t), (3) the sufficiently large supply variation caused by the good or poor harvest which is able to exert a noticeable impact on the global food market. We have done this match across all published SAMs by the Global Trade Analysis Project (GTAP) database version 10 and the best available RS-forecasting results of wheat, corn, and soybean harvests in the Global Agriculture Monitoring System of the NASA Harvest Program (GLAM; <https://glam.nasaharvest.org/>). GLAM has been led by scientists in my department at the University of Maryland, College Park. Our choice of the soybean case is further enabled by the recently published remote-sensing dataset (Song et al. 2021, "Massive soybean expansion in South America since 2000 and implications for conservation." *Nature Sustainability*). As a result, we identified the two cases for wheat, one case for soybean, but did not find a valid case for corn. **We have now added a new sub-section in Method to explain our choices of these three cases (page 8, lines 224-235).**

3) Why are you not including speculative activities in your simulation?

RESPONSE: It is because of the following two reasons. First, data limitations: Speculative activities can be challenging to quantify accurately since they involve subjective expectations and actions of individual market participants. CGE models rely on extensive data to calibrate and validate the model, and incorporating speculative activities requires additional data that are not readily available or reliable. Second, a focus on supply shocks: CGE simulations of food supply shocks typically aim to understand the direct impacts of changes in supply, such as crop failures, natural disasters, or trade disruptions. By focusing on the physical aspects of the supply shock, other factors like speculative activities, which may affect prices indirectly, might be omitted to isolate the direct effects.

As stated in the limitations of our study, speculative activities might encourage farmers' reactions, while farmers' responses might also dampen speculative activities. These factors could have a significant impact on the overall market dynamics and are worth exploring in future research.

28th Jul 23

Dear Professor Sun,

Your manuscript titled "Satellite forecasting of crop harvest can trigger cross-hemispheric production response and improve global food security" has now been seen by our reviewers, whose comments appear below. In light of their advice we are delighted to say that we are happy, in principle, to publish a suitably revised version in Communications Earth & Environment under the open access CC BY license (Creative Commons Attribution v4.0 International License).

We therefore invite you to revise your paper one last time to address the remaining concerns of our reviewers. At the same time we ask that you edit your manuscript to comply with our format requirements and to maximise the accessibility and therefore the impact of your work.

EDITORIAL REQUESTS:

*****Please take care to match our formatting and policy requirements. We will check revised manuscript and return manuscripts that do not comply. Such requests will lead to delays. *****

SUBMISSION INFORMATION:

OPEN ACCESS:

Communications Earth & Environment is a fully open access journal. Articles are made freely accessible on publication under a [CC BY license](http://creativecommons.org/licenses/by/4.0) (Creative Commons Attribution 4.0 International License). This license allows maximum dissemination and re-use of open access materials and is preferred by many research funding bodies.

For further information about article processing charges, open access funding, and advice and support from Nature Research, please visit <https://www.nature.com/commsenv/article-processing-charges>

At acceptance, you will be provided with instructions for completing this CC BY license on behalf of

all authors. This grants us the necessary permissions to publish your paper. Additionally, you will be asked to declare that all required third party permissions have been obtained, and to provide billing information in order to pay the article-processing charge (APC).

Please use the following link to submit the above items:
[link redacted]

Best regards,

Alienor Lavergne, PhD
Associate Editor
Communications Earth & Environment

REVIEWERS' COMMENTS:

Reviewer #1 (Remarks to the Author):

The author(s) addressed my comments and I satisfied with the manuscript. I recommend publication.

Reviewer #2 (Remarks to the Author):

The authors have satisfactorily addressed most of my concerns. In particular, the authors have greatly streamlined the manuscript by combining figures and moving figures/tables.

Point-by-point response to the editorial requests and reviewers' comments

EDITORIAL REQUESTS:

*****Please take care to match our formatting and policy requirements. We will check revised manuscript and return manuscripts that do not comply. Such requests will lead to delays. *****

RESPONSE: We carefully reviewed all specific editorial comments and requests regarding our manuscript in the Editorial Requests Table and outlined our response to each request in the right-hand column. The completed table is uploaded as a Related Manuscript file in this submission.

REVIEWERS' COMMENTS:

Reviewer #1 (Remarks to the Author):

The author(s) addressed my comments and I satisfied with the manuscript. I recommend publication.

Reviewer #2 (Remarks to the Author):

The authors have satisfactorily addressed most of my concerns. In particular, the authors have greatly streamlined the manuscript by combining figures and moving figures/tables.

RESPONSE: We greatly appreciate the recommendation of the two reviewers.